# The Drivers-Pressures-State-Impact-Response Model to Structure Cause-Effect Relationships between Agriculture and Aquatic Ecosystems

Alexandre Troian [1,*] , Mário Conill Gomes [1], Tales Tiecher [2], Julio Berbel [3] and Carlos Gutiérrez-Martín [3]

1   Eliseu Maciel Agronomy School, Federal University of Pelotas, Campus Universitário, s/n,
    Capão do Leão 96010-610, RS, Brazil; mconill@gmail.com
2   Faculty of Agronomy, Federal University of Rio Grande do Sul, Av. Bento Gonçalves 7712,
    Porto Alegre 91540-000, RS, Brazil; tales.tiecher@ufrgs.br
3   WEARE—Water, Environmental and Agricultural Resources Economics Research Group, Campus Rabanales,
    Universidad de Córdoba, Ctra N-IV km 396, Edificio Gregor Mendel CP, 14071 Córdoba, Spain;
    es1bevej@uco.es (J.B.); carlos.gutierrez@uco.es (C.G.-M.)
*   Correspondence: xtroian@gmail.com

**Abstract:** Different segments of society have shown interest in understanding the effects of human activities on ecosystems. To this end, the aim of this article is to analyze the scientific literature on the application of the Drivers-Pressures-State-Impact-Response (DPSIR) conceptual model to identify the parameters used to describe the causal interactions that occur between agriculture and aquatic ecosystems at the watershed scale. In this way, descriptive indicators were established for the data of 63 publications collected through Scopus, Web of Science, and Science Direct. The results confirm the great heterogeneity in the interpretation of the pressure, state, and impacts components. Part of this discrepancy can be attributed to the use of different indicators, as the model is flexible and generic. Overall, the DPSIR is a tool used not only in the scientific field, but also has demonstrated its potential to guide public policy formulation, planning, and decision-making in water resource management.

**Keywords:** conceptual framework; water management; agriculture

## 1. Introduction

Activities developed to meet human needs exert stress on the environment, and quantifying this stress is a complex task. It is of great importance to identify the mechanisms that are able to help in the organization and understanding of the causal interactions between society and the environment, in order to guide public decisions toward ensuring social welfare. In this sense, an instrument that has demonstrated potential for the structuring of complex environmental problems resulting from the processes of the interactions that occur between society and the environment is the theoretical framework of the DPSIR (driver, pressures, state, impact, and responses) model [1].

The origin of this framework results from the fusion of two fields of study, sometimes seen as opposites—ecology and economics. Short-term economic rationality and ecological cycles that ensure the renewal of nature and sustainable development do not always coincide [2]. This theoretical framework has become popular among scientists and decision makers to integrate the economic and environmental dimensions [3], and it has been used by several international reference organizations, such as the Organization for Economic Cooperation and Development (OECD), the United Nations Environment Program (UNDP), and the Environmental Protection Agency (EPA). The member states of the European Union, for example, have adopted this framework as an integrated environmental assessment strategy to support decision-making processes in the field of water resource management [4].

In academia, it is also possible to identify the use of the DPSIR framework in different approaches: in the integrated management of water resources in coastal zones [5,6], mobility and growth of urban populations [7], in the management of surface and groundwater [8–10], as a tool to support decision processes [11,12], to assess the impacts of climate change [13,14], to assess issues related to sustainable development [15,16], and for the development of environmental indicators [17–19].

Although different applications of the DPSIR structure can be identified, to the best of our knowledge, no analysis has been conducted to organize and systematize studies focusing on the socioeconomic and environmental problems derived from the development of agricultural activities, which is important to consider given that agriculture plays an elementary role in this debate, as it uses approximately 50% of the planet's habitable land and consumes more than 60% of its fresh water volume [20]. Several studies have been conducted at a watershed scale that correlate aquatic ecosystem degradation and pollution to agricultural activities [21–25]. For example, agricultural land use is one of the main factors affecting nutrient status and sedimentation in streams [26], with its effects extending to fish community composition [27]. The poor status of many aquatic ecosystems requires the restoration of catchments and improving agricultural practices [28].

In this way, through an interdisciplinary perspective, the aim of this paper is (a) to develop a bibliometric analysis of current scientific production to examine the use of the DPSIR framework in assessing interactions between agricultural activities and aquatic ecosystems. In addition, we seek to (b) point out how authors have ordered the parameters observed in the DPSIR chain, and to (c) synthesize the elementary parameters for a cause−effect analysis in watersheds. To this end, the following section describes the methodology used to assemble the database. The third section portrays the historical evolution of the structure, and lastly, the main results of this research and the considerations are presented.

## 2. Brief Evolution of the DPSIR Structure

Man's interaction with nature is as old as his existence, but in the twentieth century, the effects of this coexistence were intensified and started having global consequences. Against this backdrop, several initiatives have emerged around the environmental perspective, especially after the Stockholm Conference (1972). We highlight the conceptual framework developed by the National Statistics Office of Canada, which aimed to integrate and describe statistical data on environmental transformation under the effects of anthropogenic activities.

The indicators of the Environmental Stress Response Statistical System (S-RESS) are basically derived from data showing the s48tate of the environment (dependent variables) and data representing environmental stress (independent variables). Accordingly, Rapport and Friend [29] classified human activities, corresponding to environmental transformations, into four structuring groups, namely: (a) waste generation, (b) expansion of urbanization and agricultural areas, (c) consumption of renewable resources, and the (d) extraction of nonrenewable resources.

The understanding of the interactions between humans and ecosystems has evolved, and in the 1990s, the stress response model underwent reformulation. Initially, the chain was identified as PSR (pressures−state−response), in which pressures on the environment (including emissions or pollution) modify the state of the environment (e.g., alteration of water flow and ecosystem composition) and society responds with the aim of preventing, reducing, or mitigating environmental damage through economic, social, and environmental policies and programs [3,30].

The PSR approach was recognized by several international reference bodies; however, the information and simulation models used by the European Environment Agency (EEA) contained data not only on the pressures, state, and response, but also on their origins in economic activities. Thus, the EEA proposed a second restructuring in the causal chain [31], organized by five components arranged in cyclic systems [32] (Figure 1).

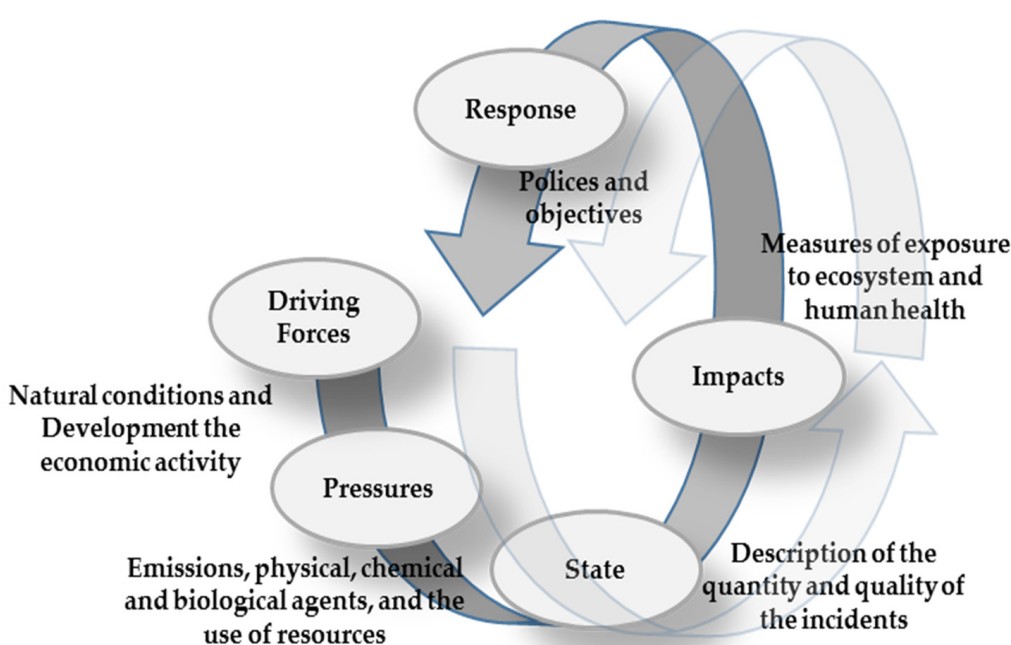

**Figure 1.** DPSIR framework: continuous feedback process. Source: [33].

The cycle starts with the inducing forces (drivers) that generate stress and cause positive or negative pressures (pressures) on the natural environment. These effects can alter the physical, chemical, and biological state (state) of the natural system, and cause impacts (impacts) on ecosystems and human health. Society usually reacts (responses) with the perspective of mitigating impacts that may affect human well-being. Responses are normally expressed in measures of the prioritization of objectives and goal setting, use of economic and legal instruments, or even through the use of technological devices [3,34]. Then, with the effects derived from the response(s) (or due to some transformation process in the driving forces of another nature), the primary cycle DPSIR becomes a new phase of pressures−state−impact−response.

Although the structure of DPSIR has been formally established, some adaptations can be identified. In a recent review of 152 articles and a list of 27 research projects related to the DPSIR framework and its derivatives in coastal and marine ecosystems, 23 variations of the model were found [35]. In general, these are adaptations used to meet specific requirements of the researchers. Therefore, this is not a formal model in the scientific field, but a structure capable of assisting in the clarification of key issues between human interactions and nature [36].

## 3. Materials and Methods

This is an exploratory study and the method followed a set of systematic bibliometric analysis processes [37]. Steps of the research are shown in Figure 2.

In the first stage, the theme of interest was defined and the concepts that best represent this theme were explored in the titles, abstracts, and keywords of the scientific documents. In the second stage, databases were consulted and 115 documents were collected from the Scopus (58), Web of Science (39), and Science Direct (18) platforms. After excluding duplicate publications, we analyzed the content of the 63 publications available in English.

In the third and last stage, the information was synthesized and the results were presented through descriptive indicators regarding the areas of knowledge, authorship, and geographical location of the authors' affiliation. In addition, the documents selected to make up the bibliometric sample were classified so as to compare the way specialists approached the possible cause−effect interactions between agriculture and aquatic ecosystems. As an analysis parameter, seven criteria were used to identify the instruments and procedures adopted (Table 1).

**Table 1.** Criteria used to classify the bibliometric sample.

| Criteria | Description |
|---|---|
| 1. Use of DPSIR | Single or in conjunction with other methodologies. |
| 2. The research approach | Interest in qualitative (analyzing attributes related to water quality), quantitative (collecting and quantifying data related to the quantity and availability of water), or mixed information (using data related to water quantity as well as in information related to water quality). |
| 3. The nature and use of the information generated | Documents that organized secondary data but did not conduct a case study, documents that used secondary data to present a case study, or documents that generated new data and information from empirical studies. |
| 4. Approach to the problem | Greater concern with the indicators, greater interest in the nature of the phenomena, or presents a balance between measuring and understanding phenomena. |
| 5. Contribution of the analysis | Exploratory (although it characterizes a problem, is more concerned with exploring and presenting the approach), descriptive (aims to describe the characteristics of a problem from the approach), and explanatory (seeks to determine the nature of the relationship between the causes and the doings of the analyzed problem). |
| 6. Collaborations between institutions for the development of research | National or international. |
| 7. Interaction with stakeholders for the development of the model | Nonparticipative analysis or with the participation of stakeholders in the problem. |

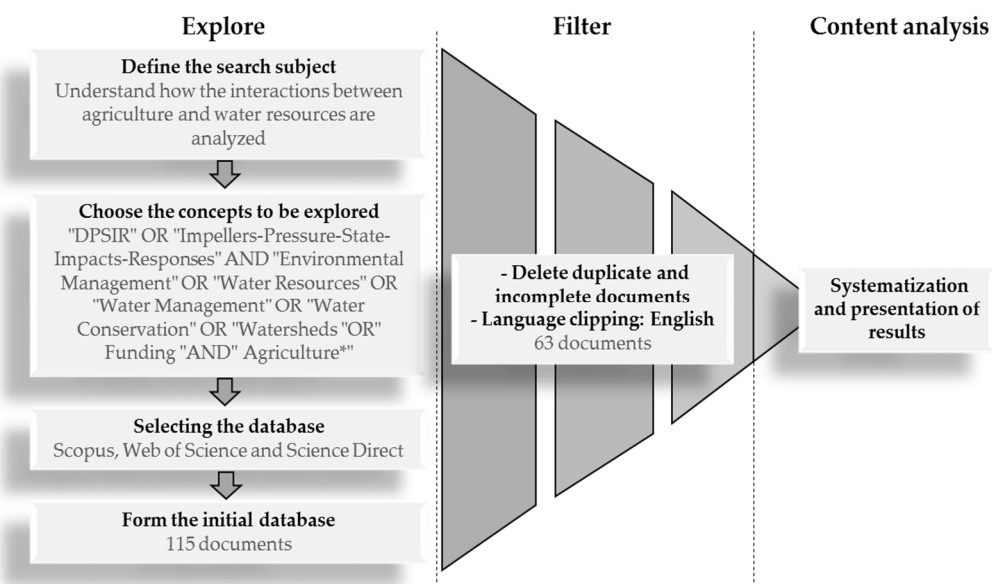

**Figure 2.** Stages of research. Source: Elaborated by the authors.

## 4. Results

This bibliometric analysis of the DPSIR conceptual framework shows that scientific production applied to the structure of the interactions between agriculture and aquatic ecosystems has been concentrated in the last two decades, totaling 63 documents since 2004. These documents were published in 37 journals, as 3 books or chapters, and as part of 3 conferences. The majority of this production is as digital articles (85.7% of the total). Although the 63 publications are distributed in 17 research areas or categories (Environmental Science; Agricultural and Biological Sciences; Social Sciences; Decision Sciences; Earth and Planetary Pciences; Engineering; Computer Science; Biochemistry, Genetics and Molecular Biology; Business, Management, and Accounting; Energy; Multidisciplinary;

Green Sustainable Science Technology; Biodiversity Conservation; Ecology; Environmental Studies; Geography; and Limnology) defined by the Scopus, Web of Science, and Science Direct platforms, 47 of them (74.8%) belong to only five categories, namely: environmental sciences (45.2%), agricultural and biological sciences (11.3%), social sciences (8.7%), decision sciences (5.2%), and Earth and planetary sciences (4.3%).

Regarding the geographical location of the first authors' affiliation (Figure 3), 66.7% were located in Europe, a large percentage were in Greece and Italy (27% of the total), and 17.5% were located on the Asian continent, especially in China in the last decade. American institutions produced 6.3% of the documents; all of the documents were from American authors. Five documents were registered in countries other than the place of affiliation of the first author; there was a group of authors with European affiliations who produced works in America, Asia, and Africa. Altogether, there were 293 authors, 14 participated in more than one document, and there were 150 scientific affiliations in 28 countries around the globe.

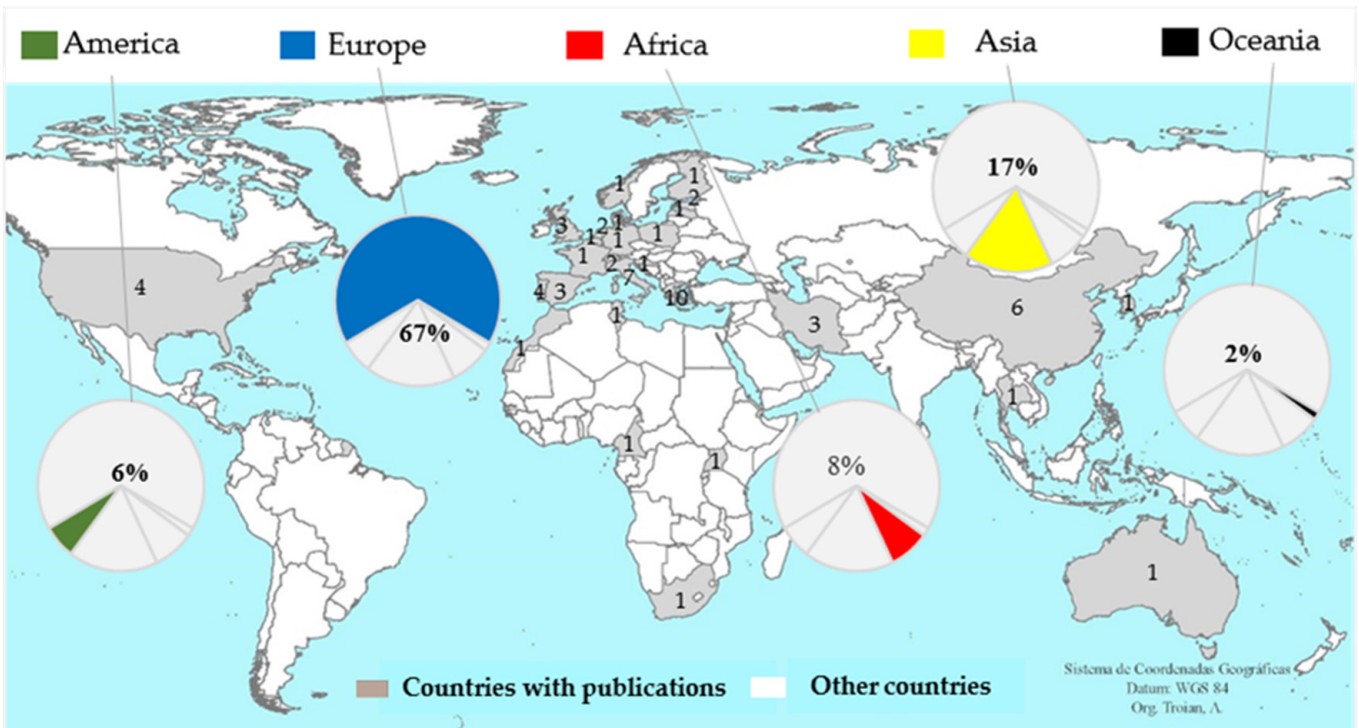

**Figure 3.** Geographic location of the authors affiliation. Source: Elaborated by the authors.

A high proportion of the publications analyzed were coauthored. More than 95.2% were published with two or more authors. Moreover, approximately 22% of the papers had four authors. In the majority of coauthored studies (78%), collaboration was at a national scale, involving institutions, research centers, or universities in the same country. In only 22.2% of cases was the collaboration at an international scale.

### 4.1. Content Check

According to the lexicographical (8-bit Unicode Transformation Format) analysis carried out on the document abstracts, 16,543 occurrences could be observed in active form, of which 997 were hapax (6% of the occurrences). Among the active forms, which excluded the presence of articles, prepositions, conjunctions, and complementary verbs, the word water was the most frequent. The occurrences were structured in 2348 text segments, organized in hierarchical classes that could be used to interpret the occurrences. This means that on average, seven active words were used to express information. Figure 4A shows the

number of occurrences and the distribution word frequency among all of the documents. Vertical axis shows word frequency in documents analyzed; while the length of segments in the horizontal axis represent the set of words found in the documents. Figure 4B shows the words arranged graphically according to their frequency (words occurring at least 20 times), the more centralized and larger the spelling, the more expressive is the frequency of words in the documents.

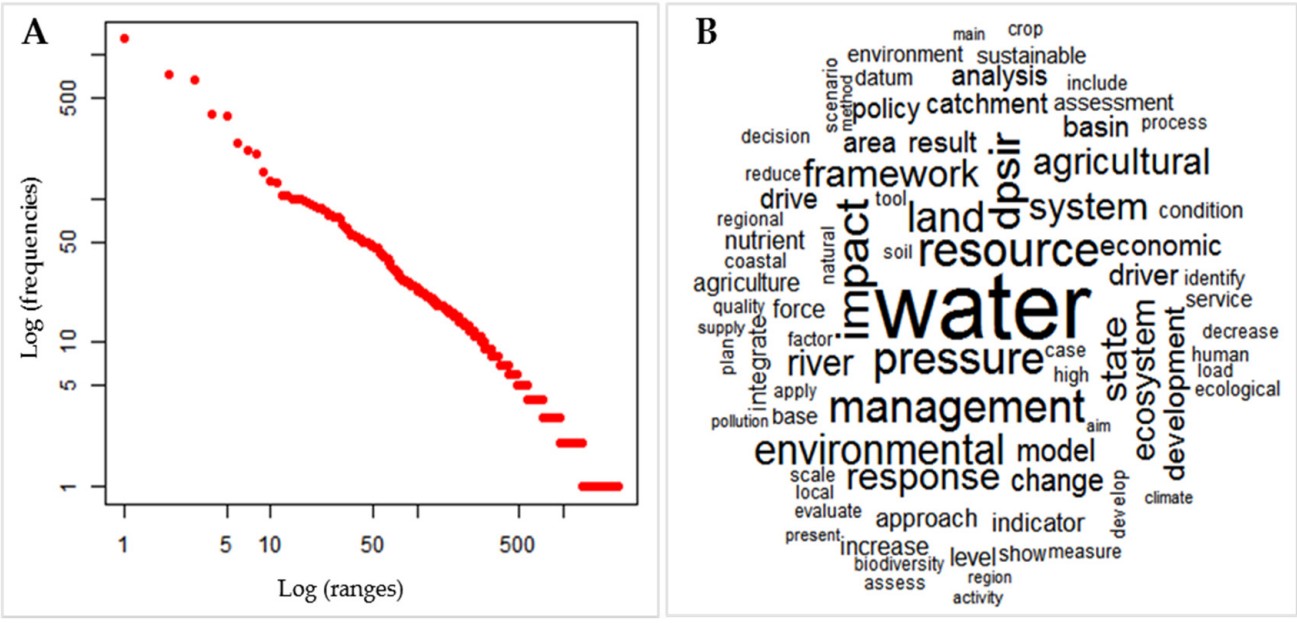

**Figure 4.** (**A**) Distribution of the frequency and (**B**) proximity in which the words appear in the summaries. Source: Elaborated by the authors.

The compiled documents were assembled into large topics (according to highlighted color), according to the purpose of the investigation. Next, they were organized according to the following criteria presented in Table 2.

1. Use of DPSIR—(A) single or (B) in conjunction with other methodologies;
2. Predominant indicators in the analysis—(A) water quality, (B) water quantity, or (C) both attributes (quality and quantity);
3. Source and use of information—(A) generate information from already prepared material and does not present a case study, (B) use secondary data in a specific case study, or (C) produce primary data and, together with secondary data, analyze a specific case study;
4. Approach to the problem—(A) concern with indicators, (B) interest in the nature of the phenomena, or (C) balance between measuring and understanding the phenomena;
5. Level of contribution of the analysis—(A) exploratory, (B) descriptive, or (C) explanatory;
6. Collaboration for the development of research—(N) national or (I) international; and
7. Participation of stakeholders—(S) participatory or non-participatory construction model.

**Table 2.** Characterization of documents.

| Reference | Object of Analysis | 1 | 2 | 3 | 4 | 5 | 6 | 7 |
|---|---|---|---|---|---|---|---|---|
| [38] | Participatory approach to water quality problems | A | C | C | A | B | I | S |
| [12] | Combination of approaches for agricultural management | B | C | B | C | A | N | |
| [39] | Agricultural management and the environment | A | B | B | C | C | N | |
| [40] | Participatory policies for socio-ecological systems | B | C | C | A | C | N | S |
| [41] | Seasonal human migration in search of water | A | B | C | B | C | I | S |
| [42] | Sustainable water management | A | C | C | C | C | I | S |
| [43] | Soil and water conservation policy | A | C | B | C | C | N | |
| [44] | Cognitive engineering in the management of water resources | A | A | C | A | B | N | S |
| [45] | Groundwater management | A | C | C | C | C | I | |
| [46] | Criteria for the use of wastewater | B | C | C | B | B | N | S |
| [47] | Scenarios for management of environmental resources | B | C | B | A | B | N | |
| [7] | Integrated management of water resources | A | C | C | B | C | N | |
| [48] | Planning for large-scale water management | B | C | C | C | B | N | S |
| [49] | Strategic planning for risk management | B | C | C | C | B | N | S |
| [50] | Management of coastal zones | A | C | B | B | C | N | |
| [51] | Management strategies, conservation, and restoration | B | C | C | A | B | I | |
| [52] | Decision support under N pressure in agriculture | B | B | B | A | A | I | |
| [53] | Transfer of agricultural nutrients (P) to water | B | A | C | B | B | N | |
| [54] | Transfer of N and P from diffuse sources (agricultural) for water | A | A | B | B | B | N | |
| [55] | Agro-environmental indicators for agricultural N monitoring | A | A | B | B | B | N | |
| [56] | Water pollution by heavy metals | A | A | C | A | C | N | |
| [57] | Industrial and agricultural nutrients (P and N) in coastal areas | B | C | C | C | C | N | S |
| [5] | Socioeconomic analysis and transfer of pollutants for water | A | A | B | A | C | N | |
| [58] | Pollution of the ecosystem based on nutrients (N and P) | A | A | C | A | B | N | |
| [59] | Urban and agricultural pressures on water resources | B | C | B | B | B | I | |
| [19] | Hierarchy of socioeconomic indicators | B | C | B | B | C | N | |
| [60] | Pressure factors in water resources | B | A | C | C | B | N | S |
| [61] | Wastewater and agricultural pressures in river pollution | A | B | B | B | B | N | |
| [62] | Socioeconomic drivers and pressures on ecosystems | A | C | C | A | C | N | |
| [63] | Implications of the driving forces in coastal areas | A | C | B | A | B | I | |
| [64] | Influence of socioeconomic change on water quality | A | A | C | B | B | N | |
| [65] | Degradation of groundwater | A | C | B | B | C | N | |
| [66] | Ecosystem health index | A | A | B | B | C | N | |
| [67] | Agri-environmental indicators of agricultural intensification | A | C | C | C | B | N | S |
| [18] | Indicators for agricultural water and land resources | A | C | B | B | A | N | S |
| [68] | Ecological status of water | A | C | B | A | B | N | |
| [69] | Loading capacity of water and land resources | B | C | B | A | B | N | |
| [70] | Environmental status when implementing CAP measures | B | C | B | A | A | N | |
| [71] | Change in land use and ecosystem services | A | C | C | A | C | N | S |
| [72] | Changes in land use and pressures on water | A | B | B | A | C | N | |
| [73] | Environmental impacts from land use change | B | C | B | B | C | N | S |
| [74] | Change of land and the consequences on soil and water | A | C | B | A | B | N | |
| [75] | Change in land use, conflicts or synergies | B | A | A | A | C | I | S |
| [76] | Impact of land use change | A | C | B | A | A | N | |
| [77] | Conceptual model for water resources and climate change | B | C | B | C | C | N | |
| [78] | GIS to assess pressures on water resources | A | C | B | B | C | N | |
| [79] | Conceptual model for socio-ecological research | B | C | B | A | C | N | |
| [80] | Development scenarios in the marine environment | B | C | B | B | C | N | |
| [81] | Software to simulate impacts of climate change | B | C | B | A | C | I | |
| [82] | Model to detect agricultural diffuse pollution | B | C | B | B | B | N | |
| [83] | Changes in ecosystem services | B | A | C | A | C | I | S |
| [84] | Mechanisms of interaction in ecosystem services | A | A | C | C | C | N | |
| [85] | Socioeconomic influences on ecosystem services | A | C | B | B | C | N | |
| [86] | Compensations of ecosystem services | A | A | B | C | B | N | S |
| [85] | Ecosystem services and wetland changes | A | A | C | C | B | N | S |
| [10] | Agrarian economy in deficit irrigation | A | B | C | B | B | N | S |
| [87] | Cost-effectiveness analysis in the Water Framework Directive | B | B | B | B | C | N | |
| [88] | Agrarian economy in irrigation of mature basins | A | B | B | B | B | N | |
| [89] | Export costs of agricultural nutrients | A | A | C | B | B | I | |
| [14] | Impact of climate change on agriculture | B | B | B | B | B | N | |
| [13] | Regional climate change disturbances | B | C | C | C | C | N | S |
| [15] | Border of sustainable development with the DPSIR | A | A | A | C | C | N | |
| [90] | Sustainability in industrialized and developing countries | A | C | A | B | A | N | |

Although in most documents the DPSIR framework was used as a discrete tool, a considerable proportion of the researchers used it in combination with another approach. This was the case for the use of multicriteria decision support methods applied to test

alternatives to reduce agricultural nitrogen pressure on European water resources [52], so as to quantify ecosystem service offsets [86], to analyze surface and groundwater management scenarios [47], to structure decision problems related to water management in watersheds under agricultural pressure [12], to establish sustainability indicators [19], and to assess environmental impacts related to land use change [73].

The DPSIR framework is also combined with the use of fuzzy systems in cognitive mapping [40,48] and the strategic planning of stakeholders with a focus on risk management [49].

In addition to using DPSIR combined with other instruments, a recurring practice observed in the documents was the modeling of prospective scenarios. Modeling was used in 33% of the analyzed documents. The simulations were conditioned and based on assumed alternatives to identify a comparative advantage among them. In practice, simulations were used to guide actions and decisions toward the goals pursued in policy formulation, planning, and decision-making.

In 20 of the 63 documents, researchers actively involved experts or citizens interested in the decisions to take part in the research. However, where stakeholders were involved, few papers clearly indicated who was targeted, the number of stakeholders, and how involvement occurred. Therefore, it seems that this procedure was underutilized, as only in 32% of the documents were stakeholders' preferences for the development of the model incorporated. The results show that there is a predisposition by researchers to use quantitative and qualitative indicators concurrently. Additionally, the information used to structure the DPSIR cause−effect interaction chain originated from three main sources: literature review, secondary data from official sources, and primary data produced in the research. Although the use of secondary data was significant, over 40% of the papers produced primary data and only three papers were not case studies (Table 3).

**Table 3.** Synthesis of the characterization of documents.

| Coefficients | Procedures | Cases Identified | % |
|---|---|---|---|
| Use of DPSIR | Only | 36 | 57.1 |
| | In conjunction with other methodologies | 27 | 42.9 |
| Predominant indicators in the analysis | Water quality | 17 | 27.0 |
| | Water amount | 9 | 14.3 |
| | Both attributes (quality and quantity) | 37 | 58.7 |
| Source and use of information | Generate information from material already elaborated and do not present a case study | 3 | 4.8 |
| | Use secondary data in a concrete case study | 34 | 54.0 |
| | Produce primary data and, in conjunction with secondary data, analyze a specific case study | 26 | 41.3 |
| Focus on the problem | Greater concern with indicators | 22 | 34.9 |
| | Greater interest in the nature of the phenomena | 25 | 39.7 |
| | Balance between measuring and understanding phenomena | 16 | 25.4 |
| Contribution level of the analysis in the model DPSIR | Present a problem and relate it to the model (exploratory) | 8 | 12.7 |
| | Describe the problem and relate it to the model (descriptive) | 27 | 42.9 |
| | Identify the factors that determine or contribute to the occurrence of the phenomena (explanatory) | 28 | 44.4 |

## 4.2. DPSIR Structure Parameters and Components

The DPSIR conceptual framework provides an overview of the main environmental problems; therefore, the identification of the components and the definition of the parameters may lead to different decisions in structuring the causal chain. For this reason, some terminological differences were identified in the definition or ordering of the indicators of the DPSIR framework.

Frequently, the following indications prevailed among the driving forces: population growth (in 13 documents), urbanization (5 documents), and industrialization (7 documents). In addition, at least 23 of the documents highlighted agriculture as a driving force, while 7 of them highlighted animal husbandry (Table 4).

**Table 4.** Definitions of DPSIR categories in agrarian systems.

| Reference | Factors | | | | | | | |
|---|---|---|---|---|---|---|---|---|
| | Agricultural | Livestock | Fertilizers | Change in Land Use | Water Extraction | Nutrients/ Contaminants | Amount/ Quality Water | Eutrophication |
| [65] | D | | | P | P | | S | |
| [12] | D | | P | | P | I | S | |
| [53] | D | P | P | | | S | | I |
| [38] | D | | P | | | S | | S |
| [40] | P | | | | D | | S | |
| [41] | D | D | | D | P | | I | |
| [10] | | | | | P | | S | |
| [71] | P | | | | P | | S | |
| [39] | | | P | | P | | S | |
| [19] | | | P | | P | | I | |
| [42] | D | D | | I | P | S | S | |
| [83] | D | | P | | | S | | |
| [43] | D | P | | P | | | | |
| [67] * | D | | P | | I/S | I/S | I/S | |
| [82] | P | | S | S | P | | S | |
| [60] | | | | I | P | S | P | |
| [78] | D | | | | P | | S | I |
| [88] | | | | | | | | S |
| [85] | | P | | P | | S | | |
| [68] | D | D | P | P | | S | I | I |
| [47] | D | | | | P | | I | |
| [8] | D | P | P | P | | S | I | I |
| [44] * | D | D | | P | P | | S/P | |
| [69] | | | | I | S | | | |
| [48] | I | | | | | P | | S |
| [73] | D | | | D | P | P | S | |
| [81] | D | | | | S | S | I | |
| [51] | D | D | P | P | P | S | S | I |
| [11] | D | P | | | | | | |
| [57] | D | | | | P | P | S | I |
| [5] | D | D | P | | | I | S | S |
| [52] | D | D | P | | | S | I | I |
| [64] | D | | | | P | P | | I |
| [76] | D | | P | | | S | | I |

Note: (D) drivers, (P) pressures, (S) state, (I) impacts, (R) responses, and (*) combined use of components.

Among the authors who included land-use change as a parameter, most classified it in the pressures category (in 8 documents); in these cases, population density and agriculture formed the driving forces. Land use change was analyzed for a wide range of possibilities, the most frequent ones being based on indicators of the intensification of production and crops in mountain environments, which in turn cause increased water extraction rates, deforestation, and changes in biotic and abiotic landscape conditions.

For example, in 16 cases, the authors of the compiled documents allocated water extraction as a pressures component. Overexploitation, water management, and water demand were the main indicators used. On the other hand, in five documents, water extraction was specified as a state, impact, or driving force component.

In one scenario, water extraction was classified into two categories simultaneously: impact and state. In this case, excessive water extraction from the Sarno River ($10,515 \times 10^6$ m$^3$ y$^{-1}$) aggravated the water resource conditions and caused changes in the state of the river. On the other hand, water extraction results in large domestic and industrial effluent discharges from that region of Italy, which worsened qualitative indicators such as the concentrations of nitrite, nitrate, phosphorus, metal, and the chemical oxygen demand. Therefore, it may also have a form of impact on aquatic ecosystems [61].

Another aspect that needs to be clarified is the use of fertilizer. Although fertilizers and nutrients are often considered synonyms [64], in the DPSIR chain they can be evaluated differently. One mode of evaluation is when fertilizers used in agriculture are observed with regard to dosage and application techniques; another is when they are observed based on the presence of residues found in water as mineral or organic nutrients. In the first case, in view of the DPSIR proposal, fertilizers are considered to be substances that can put pressure on the environment and water resources, while in the second case, the presence of phosphates, nitrites, and nitrates in the water are indicators of the natural state of change in water resources.

In addition, some authors have assigned water contamination due to the presence of nutrients, suspended solids, pesticides, and other toxic substances to the pressures component. Another group of authors assigned the different forms of water pollution to the impact component. In summary, fertilizers, nutrients, and pollutants can easily be fitted to any of the components, including: state, pressures, and impact. For example, nitrogen could be considered a driving force when mistakenly applied to crops; when evaluating the process of erosion and leaching, nitrogen concentrations would become part of a state change; and it can be considered a pressure when assessing the efficiency of its use on crops [52].

Water quantity and quality parameters were used in combination and sometimes under separate conditions. Although the quantity and quality parameters were represented by a change of state in 15 documents, impact in 8 and pressures in 2, the indicators found in this set of documents were similar.

Water balance, decreased availability, nutrient concentration, presence of toxic components, and salinization of water stand out as state indicators [39,51]. Among the documents that considered impact, the main indicators used were dissolved oxygen, electrical conductivity, hydrogen potential (pH), increased biochemical oxygen demand, and chlorophyll concentration [8,68]. Finally, water scarcity and droughts, which characterize the reduction in the amount of water available, were among the pressure indicators.

Eutrophication of water resources was one of the parameters considered in 13 documents. Nine of them established this phenomenon as an impact, because they considered the eutrophication process to be an event triggered by factors that "exert pressure" on water resources. Eutrophication was considered as a change in the conservation status of water resources in only four documents. In some cases, eutrophication was used as an indicator of water quality; in others, it was synonymous or complementary to water quality, measured through the concentration of chlorophyll, nitrogen, and phosphorus.

Although in the compiled documents there were different views on the dividing line that defined the driving forces, the greatest divergences focused on the definitions of what constitutes pressures, impact, and state.

With respect to responses, a wide variety of actions and policies have been verified, ranging from monitoring to technical, regulatory, and subsidy provisions to prevent, compensate, or adjust possible changes in the state of the environment, and may be directed at individuals or public or private social groups. Among the actions verified, the most common were water use restrictions, water and watershed management plans, economic mechanisms related to water prices and tariffs on pollutants, technical and financial incentives to modernize irrigation systems, training programs for water users, etc. No references were made to wastewater treatment by small towns or rural populations, which is a recurrent problem in river basin studies [22].

Although the answers mentioned are apparently consistent, the implementation of actions has been minimal. In addition, there is a predominance of global or national responses. Local or social group responses are virtually nonexistent. In fact, the responses are presented as suggestions; they do not express objective proposals, with timelines, implementation costs, or anything similar, that technically support the suggested arguments.

## 5. Discussion

In general, while there is consensus on the use of spatial and temporal indicators to identify an anthropogenic effect on the environment, the distinction between pressures, state, and impact is not uniform among authors. There are cases in which impact is described differently by social and natural scientists, as negative changes in human well-being, in contrast with unfavorable changes in the natural environment, respectively.

Analogous ambiguities are also found for other references. For example, in a review of 24 papers aiming to assess the potential of the DPSIR framework in coastal social-ecological systems, the authors indicated that there was consistency in the formulation of the parameters that make up a driving force; however, they highlighted that there was little consensus on the definitions of pressures and impact [1].

In another review, the authors concluded that the extensive diversity of terminologies used among scientists from different fields of knowledge contributes to the discrepancies in definitions regarding the categories of the DPSIR framework. There are cases where driving forces are subdivided into primary and secondary or into underlying and immediate, and other examples refer to driving forces as physical and socioeconomic; natural and anthropogenic, indirect, and direct, endogenous and exogenous, etc. [91].

According to our systematization presented in Figure 5, first, driving forces are defined by the coexistence of genuine natural factors and factors associated with the development of human activities. In the first case, the natural environment is highlighted in the form of climate, geology, and topography, which determine the underlying character of watersheds. These factors generally have a positive influence by ensuring the renewal of natural resources, while anthropogenic interference tends to be predominantly of a negative character.

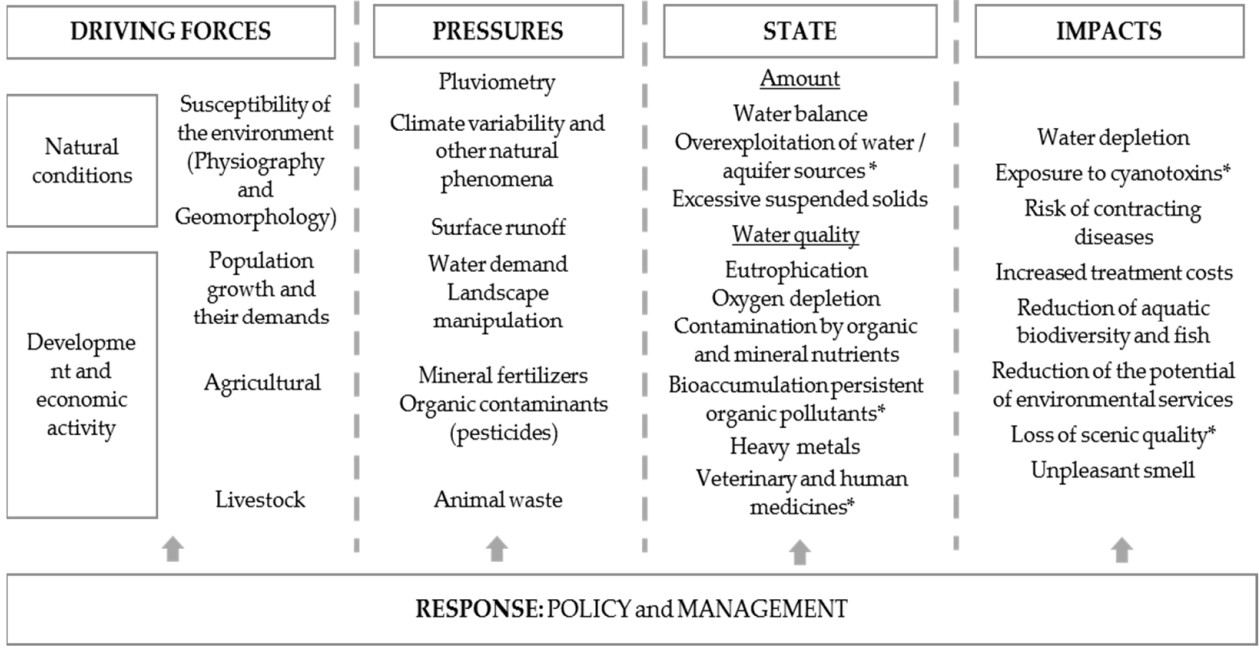

**Figure 5.** Elements of the DPSIR chain for agriculture and water resources. Note: (*) parameters added to the analysis chain not mentioned in the literature consulted. Source: [33].

Anthropogenic interference in the natural environment generally occurs as a result of population growth and increased demand for resources required to meet growing human needs. Among the activities indispensable for the development of humanity, agriculture, which involves both plant crops and animal husbandry, is considered one of the main driving forces of this system of interactions.

Similarly, there are pressures derived from the natural environment, in particular variations in the rainfall regime, such as changes caused by extreme events such as "El Niño" and "La Niña", which occur with great intensity in Amazonia and Northeast Brazil and in some regions of Australia and Indonesia. Other phenomena derived from the natural environment, which can also put pressure on the balance of ecosystems, are hurricanes, snowfalls, volcanic eruptions, tsunamis, earthquakes, etc.

On the other hand, anthropogenic pressures are generally characterized by the description of how natural resources are used and the amount of waste transported and deposited in the environment, especially in aquatic systems. Changes in the landscape, which include land use and natural resource management, in the case of agriculture, are among the main drivers of anthropogenic pressures in our assessment. In this case, land use can be analyzed against a wide range of possibilities, the most common ones are based on indicators of crop intensification in environments with a low agricultural potential, deforestation, and changes in biotic and abiotic conditions of the landscape. The inputs used in agricultural production systems are also part of the set of pressures to be observed, especially pesticides and fertilizers, as these are not fully absorbed in the production processes and residues that are harmful to the environment accumulate.

Situations in which ecosystems are considered in relation to the conditions of conservation or degradation are evaluated on the basis indicators related to the state of the natural environment (physical, chemical, and biological state). According to our approach, these indicators should basically portray quantitative information regarding the water balance and identify the presence of toxic components in watersheds (qualitative and quantitative data).

Impacts are characterized by indirect and diffuse disturbances, i.e., their indicators generally include measures of the health of natural systems (animal morbidity, extinction of species, etc.) and measures of human well-being (diseases transmitted by microorganisms, fraction of the population with access to water below quality standards, etc.). Contingencies that affect the economy, such as environmental investment costs, compensation costs for environmental services, provisioning, and regulation, can be considered reverse impact measures [3,31,32].

As a result, society generally reacts with the fundamental aim of mediating the observed conflicts between agriculture and ecosystems. Society's responses aim to mitigate the state of environmental degradation provided by agriculture and the impacts that may affect ecosystems and human well-being. Above all, the current challenge with regard to responses should be to overcome and transcend conflicts to exploit the potentials and synergies arising from the interaction of agriculture with its environment.

In summary, the aim of this bibliometric survey was not to organize agri-environmental indicators, but it has been acknowledged that these indicators should measure the scalar, temporal, and multidimensional effects. In our study, we observed difficulties in integrating indicators of plural dimensions. Moreover, most of them described outcomes on a global scale or one that did not consider local knowledge. Existing inequalities between actors and stakeholders have often been ignored, which minimizes the relevance of social diversity across most fields of research [15].

Another aspect that deserves emphasis in this approach concerns the rich diversity of habitats, the distinct threats to these habitats, and the great variety of conservation challenges that aquatic ecosystems face [28]. Hydrological flows form a broad and continuous river system along which sediments and dissolved agricultural substances disperse. Although interconnected, aquatic ecosystems can be divided into marine and freshwater systems. Marine systems, located in coastal zones, tend to absorb disturbances from urbanized; highly populated; and, generally, industrialized regions. In turn, freshwater regions present a more diverse mosaic with respect to human occupation patterns and land use.

In addition to the differences in use and occupation, different aquatic ecosystems respond differently to driving forces, and consequently to impacts, and therefore have different habitat restoration and recovery needs. In freshwater watersheds, impacts can

be scaled locally, and there is more scope for active restoration work. On the other hand, in coastal marine areas, contaminating pollutants may present cumulative impacts, and there is a greater dispersion and connectivity of species; consequently, there is a need to consider regional scales and a better possibility of taking advantage of the natural recovery of systems [28]. In this sense, we highlight the need to rethink the structure and functioning of these ecosystems.

Finally, the DPSIR framework has contributed to organizing and describing the anthropogenic interactions with the environment; however, it is not categorical to minimize contemporary environmental problems. Operationally, the DPSIR has functioned as a support tool, giving a medium- to long-term vision; therefore, cross-cutting issues, which refer to human capital, health, education, and gender, which have indirect relationships with environmental issues, need to be incorporated into the current concerns.

## 6. Conclusions

This literature review presents documents with applications in different scientific fields in the social and environmental dimensions, which illustrates the flexibility of the DPSIR framework to establish causal relationships between agriculture and aquatic ecosystems.

We highlight the employment of the DPSIR framework to identify and evaluate pollutants of an agricultural origin in water; examine the environmental status of water resource stressors; analyze changes in the use of land and climatic changes; qualify ecosystemic services; address sustainable development; and to manage and make decisions. Overall, this analysis concluded the following:

- The DPSIR framework demonstrates the capacity to organize and present causal relationships between agricultural activities and the environment related to ecological, social, or economic perspectives.
- DPSIR is a simple and generic application model; however, the interpretations of the variable components of pressures, state, and impact are not homogeneous. Thus, it is difficult to establish a standard of socioeconomic and agri-environmental indicators.
- In the documents analyzed here, the DPSIR model was not used to explain synergic situations between the environment and agricultural activities, that is, to present sustainable development scenarios. In contrast, they have been directed to illustrate situations where agricultural activities lead to environmental degradation.
- The stress factors of an anthropogenic origin that affect ecosystems are difficult to measure, and the available data are often limited.

In summary, this bibliometric survey demonstrated that the DPSIR approach has virtues to promote a dialog between different scientific disciplines with respect to complex environmental problems. In this sense, based on the results of the literature review, it is suggested to establish a research agenda among scientists and experts interested in this instrument to equalize the establishment and employment of indicators related to agricultural externalities. For managers, it is recommended to adopt the guidelines of the DPSIR approach as an integrated and participatory strategy for environmental assessment to support decision-making processes in the field of water resources at the watershed scale.

**Author Contributions:** Conceptualization, J.B.; methodology, A.T.; software, A.T.; validation, M.C.G., C.G.-M. and J.B.; formal analysis, T.T.; investigation, A.T.; resources, J.B., C.G.-M. and M.C.G.; data curation, A.T.; writing original draft preparation, J.B. and A.T.; writing review and editing, A.T., C.G.-M. and T.T.; visualization, A.T. and C.G.-M.; supervision, J.B. and M.C.G.; project administration, J.B. and M.C.G.; funding acquisition, M.C.G., A.T. and J.B. All authors have read and agreed to the published version of the manuscript.

**Funding:** Coordenação de Aperfeiçoamento de Pessoal de Nível Superior (CAPES) funded this research (grant number PDSE 88881.189405/2018-01) for Alexandre Troian, as well as the Andalusian Department of Economy and Knowledge and the European Regional Development Fund (ERDF) through the research project SEKECO (UCO-1263831-R).

**Data Availability Statement:** The data were organized from the following websites: Scopus, Web of Science, and Science Direct.

**Acknowledgments:** Department of Economics and FinanceUniversity of Cordoba and University of Pelotas.

**Conflicts of Interest:** The authors declare no conflict of interest.

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
