# Peer review of "The Drivers-Pressures-State-Impact-Response Model to Structure Cause−Effect Relationships between Agriculture and Aquatic Ecosystems"

_sustainability, doi:10.3390/su13169365_

Round 1
Reviewer 1 Report
The manuscript is a relatively systematic literature review of 63 publications related to the use of the DPSIR framework in analyzing the impact of water use in agriculture operations. The DPSIR framework and the research methodology are well described, however the manuscript is weak analytically and requires more work both in presenting the results and in discussing and interpreting them. In addition, the manuscript requires a serious English language editing to make it understandable to readers and to eliminate repeated words. Some of the aspects that require the authors’ attention are:
- On line 132, better explain what are the 7 criteria used to classify the bibliometric sample. You may use a table to organize them.
- On line 160, what are the 17 sub-areas of knowledge and explain how did you select the 5 categories.
- Starting on page 194, Table 1 is hard to read and understand. You may move the note that now follows Table 1 in the body of the article, explaining the structure of the table before the table.
- In Table 3, starting on line 247, what does E represent?
- The whole discussion section needs to be rewritten. At this point, Figure 5 does not explain the cause-effect interactions between agriculture and aquatic ecosystems at the watershed scale. The DPSIR framework does not make much sense without a proper context of the case study. Provide more context, and show how Responses have been developed to match the DPIs. Define agriculture and aquatic ecosystems as they appear in the publications reviewed.
- Give sources for Figures 1 and 2 on lines 100 and 122.
- Lines 348-350 do not seem to belong in the manuscript.
Author Response
We would like to thank the reviewer for the detailed comments and suggestions provided. The file with the suggested changes is attached.

Reviewer 2 Report
This paper provides a review on a drivers-pressures-state-impact-response modelt he interrelation between agriculture and aquatic ecosystems. Overall, this is an important topic and I think it is generally useful to have such a review. Here are my suggestions:
First, I think that the introduction, particularly line 59 ff would benefit from one or two concrete examples of how agriculture can affect aquatic ecosystems so that the readers have a specific example idea about the relevance. This could e.g. be a reference to papers that show that agricultural landuse is one of the main factors affecting nutrient status and sedimentation in streams (Knott et al., 2019), with effects even extending to fish community composition (Bierschenk et al., 2019), and a reference to the fact that the poor status of many aquatic ecosystems requires restoration of catchments including their agricultural practices (Geist & Hawkins, 2016).
Second, I think that the apper would benefit from another conceptual graphics that shows the identified interlinkages between agriculture and aquatic ecosystems both, distinguishing differences between marine and freshwater systems (where e.g. different nutrients and effects play a role).
Third, in terms of the conclusion, I would recommend making a clear and pragmatic recommendation on what is needed to move forward. Whilst the theoretical aspect is already well covered, an additional consideration of the useful practical next steps based on the findings of the review would be useful.
References:
Bierschenk AM et al. (2019) Impact of catchment land use on fish community composition in the headwater areas of Elbe, Danube and Main. Science of The Total Environment 652; 66-74. DOI: 10.1016/j.scitotenv.2018.10.218
Geist J, Hawkins SJ (2016) Habitat recovery and restoration in aquatic ecosystems: Current progress and future challenges. Aquatic Conservation: Marine and Freshwater Ecosystems 26; 942–962
Knott J et al. J (2019) Effectiveness of catchment erosion protection measures and scale-dependent response of stream biota. Hydrobiologia 830; 77-92. DOI: 10.1007/s10750-018-3856-9
Author Response

(The authors gave the same response as above.)

Round 2
Reviewer 1 Report
Thank you for revising the manuscript according to suggestions.